# Erosion of Granite Red Soil Slope and Processes of Subsurface Flow Generation, Prediction, and Simulation

**DOI:** 10.3390/ijerph20032104

**Published:** 2023-01-24

**Authors:** Yichun Ma, Zhongwu Li, Liang Tian, Yifan Yang, Wenqing Li, Zijuan He, Xiaodong Nie, Yaojun Liu

**Affiliations:** 1College of Geographic Science, Hunan Normal University, Changsha 410081, China; 2Jiangxi Provincial Key Laboratory of Soil Erosion and Prevention, Nanchang 330029, China

**Keywords:** granite red soil, rainfall redistribution, soil erosion, surface–subsurface flow, flow prediction

## Abstract

A deeper understanding of the rainfall–flow processes can improve the knowledge of the rain-driven erosional processes in coarse-textured agricultural soil. In this study, on the red soil slope farmland developed from weathered granite, a simulated rainfall experiment was conducted to study the characteristics of rainfall redistribution, the processes of surface–subsurface flow generation and prediction, and sediment production. Rainfalls with three intensities of 45, 90, and 135 mm h^−1^ with a duration of 90 min were applied to the weathered granite red soil with the slope gradient of 10°. Under 45 mm h^−1^ rainfall intensity, the output of rainwater was composed by subsurface flow and bottom penetration, accounting for 35.80% and 39.01% of total rainfall, respectively. When the rainfall intensities increased to 90 and 135 mm h^−1^, the surface flow became the main output of rainwater, accounting for 83.94% and 92.42%, respectively. Coarsened soil exhibited strong infiltration-promoting but poor water-storage capacities under light rainfalls. With an increased rainfall intensity, the surface flow coefficient increased from 19.87% to 92.42%, while the amount of subsurface flow and bottom penetration decreased by 1.3 and 6.2 L, respectively. For sediment production, the sediment concentration was raised from 1.39 to 7.70 g L^−1^, and D10, D50, and D90 increased by 1.50, 1.83, and 1.40 times, respectively. The content of coarse particles (>1 mm) in surface soil increased by 12%, while the content of fine particles (<0.5 mm) decreased by 9.6%. Under strong rainfalls, severe soil and water loss, coarsening soil surface, and large loss of fine particles became major problems. During rainfall, the subsurface flow and bottom penetration could be predicted well through quadratic equations of rainfall time, which transformed into time-dependent exponential functions after rainfall. The results provide a theoretical basis and data reference for soil erosion prevention and water management in coarse-textured agricultural lands.

## 1. Introduction

Rainfall-driven soil erosion decreases soil fertility and productivity, threatens the ecological environment, and limits agricultural production [1]. More and worse soil erosion are primarily related to rainfall events [2,3], especially the frequent occurrence of climate change and extreme rainfall events in recent years [4].

The soil developed from the weathered granite materials has a high content of sand, which results in coarse soil texture, strong infiltration rate, poor water storage capacity, and anti-erodibility [5]. Therefore, this area is prone to severe soil erosion under heavy rain [6,7]. In addition, the weathered granite soil has a distinct layered structure characteristic of “upper-soil, lower-sand, and bottom-rock”. Such soil properties are conductive to infiltration and the development of subsurface flow in the soil body [8,9]. Subsurface flow affects the distribution of rainfall–flow and the soil erosion process through direct ways such as infiltration and pipe flow or indirect ways such as changing soil properties and increasing water pressure [10,11,12,13]. Bryan and Rockwell [14] found a significant increase in sediment transport under the effect of subsurface flow. Huang et al. [15] pointed out that the soil erosion enhanced significantly after the subsurface flow occurred. The existence of subsurface flow could weaken the cementation, cohesion, and internal friction angle among soil particles, then affect the soil strength and slope stability, leading to collapse, slump, and debris flow disasters [16]. Li et al. [4] found that landslides could be triggered after the subsurface flow in weathered granite areas, reaching a critical value.

The red soil hilly region of southern China is an important agricultural production area [17], accounting for 36% of the total cultivated land and more than 50% of the total output of oil-bearing crops, grain, and agricultural in China [18]. The red soil sloping farmland developed from the weathered granite parent materials is widely distributed here and is the primary source of soil erosion [19,20], and the soil loss rate can reach 98.78 t ha^−1^ yr^−1^. There is a large total amount of rainfall in this area but with extremely uneven distribution of temporality and spatiality. More than 70% of the rainfall concentrates in March to July [21,22], mainly including two rainfall types. In spring, plum rain with small intensity lasts for a long duration, causing a large, accumulated amount. In summer, thunder showers and short-term rainstorms become more frequent; the duration is shorter but with heavy intensity, which also results in high total rainfall amount. Different rainfall types alter the rainfall–flow redistribution and the processes of flow generation and soil loss [23,24,25]. Due to different rainy seasons with high precipitation, hilly terrain, and coarse soil texture, the red soil region has been facing severe soil and water loss [26,27,28].

Special soil properties and complex hydrological conditions lead to the processes of flow generation and soil erosion in the weathered granite red soil area that are significantly different from other regions, such as the black soil area in the northeast [29], the loess area in the northwest [30,31], the purple soil area in the southwest [32,33], and the karst area [34]. Many scholars have carried out research on the characteristics, mechanisms, and control methods of soil erosion on weathered granite [35,36]. Deng et al. [37] investigated the effects of rainfall intensity and slope gradient on runoff and sediment yield from hillslopes with weathered granite. Liu et al. [38] studied the processes and mechanisms of collapsing erosion for granite residual soil in southern China.

In the area of weathered granite red soil, the subsurface hydrological process is complex and accounts for a high proportion in rainfall–flow. The nutrients and pollutants may migrate downward with the subsurface flow. In addition, the amount of soil erosion caused by subsurface flow is much higher than that of splash erosion and sheet erosion. Under the condition of long-term high soil water content, the weathered granite slopes are prone to cause soil instability, and then more severe disasters occur such as collapses and landsides. However, the current research on soil erosion on weathered granite red soil area mainly focuses on the surface hydrological process and is insufficient for the research on infiltration and near-surface hydrological process. The mechanism of sloping soil erosion has not been fully clarified and also there is a lack of effective predictable and preventive measures. Severe soil erosion and downward water loss have seriously limited the agricultural development in the granite red soil area. More research on the characteristics of surface–subsurface flow generation is needed in this region [24].

Due to the complex process of soil erosion in weathered granite slopes, under the influence of the spatio-temporal variability of natural rainfall and the heterogeneity of soil physio-chemical properties in the wild, effective quantitative and qualitative research cannot be carried out. Indoor flow plot experiments under artificially simulated rainfall can be flexibly and simply operated with the specific conditions of typical study areas. It has become one of the most common methods in studying rainfall–driven erosion on sloping land at small and medium scales [24,39].

On the basis of the above information, the objectives of this study were to (1) analyze the redistribution of rainfall–flow on the surface; (2) study the characteristics of flow generation of different soil layers and sediment production under different rainfall intensities; and (3) predict the processes of subsurface flow and bottom penetration generation. The research results can provide a data basis and theoretical support for the management of soil erosion and rainwater resource of agricultural soil in the red soil hilly area of southern China.

## 2. Materials and Methods

### 2.1. Study Area

Soil samples were collected in Ningdu County (115°40′20″–116°17′15″ E, 26°05′18″–27°08′13″ N), which is located in the middle and low mountainous hilly area in southeastern Jiangxi Province, China (Figure 1). The study area experiences a humid mid-subtropical monsoon climate, with a mean annual precipitation of 1706 mm, and with 40–70% of the total precipitation mainly occurring from April to June. The average annual temperature is 14–19 °C, with an annual sunshine hour of 1938.8 h. The landscape of the area is dominated by hills and mountains. The rocks are mostly granite, and the red soil formed by the weathering of granite is widely distributed here. The whole soil layer has a large amount of quartz sand and gravel, with rough texture, low fertility, and strong permeability, being a serious soil erosion area in Jiangxi Province. The topography and soil conditions are typical of the granite region of southeastern China.

### 2.2. Plot Installation

The steel structure soil plot used in this study was with a full size of 1.5 × 0.5 × 0.5 m (length × width × depth) (Figure 2). According to the field observations, the plot was set at a fixed slope of 10°, which was representative of the study area. The sampling soil was collected from the surface layer 0–30 cm deep in a Masson pine forest, which had been eroded and degraded for many years. The soil texture belongs to the loamy sandy soil, with a sand content of 72.55%, silt content of 12.38%, and clay content of 15.07%. Detailed soil information is shown in Table 1.

The sampling soils were bagged after air-drying, crushing, and passing through a 5 mm sieve. The soils were filled into the plot in layers of 5 cm, and the soil layer was compacted after each filling. The total thickness of the fill was 45 cm, and the soil surface was 5 cm from the top of the plot. According to the field survey at the sampling site, the filled soil had a bulk density of 1.20 g cm^−3^. In addition, the boundaries between the soils and soil plot were forcefully compacted to reduce the marginal effect.

As shown in Figure 2, the surface flow and sediment samples were converged through the “V-shape” outlets at the front part of the soil flume, which was linked with the collection device. At the bottom of the soil plot, the water outlet was set as a square of 5 cm to ensure free infiltration during the rainfall. Bottom penetration water was collected through the splicing plate. A 10 cm wide slit was opened under the front baffle of the soil plot, and a gauze was fixed with glue inside of slit to ensure soil particles would not leak out during the soil-filling and rainfall processes. A water pipe with a valve was installed outside of slit to lead out subsurface flow, which was connected with the bucket to collect subsurface flow.

### 2.3. Rainfall Simulation

The experiments were carried out in the artificial simulated rainfall hall of Jiangxi Ecological Science and Technology Park of Soil and Water Conservation (115°42′08′′–115°43′06′′ E, 29°16′37′′–29°17′40′′ N), which was located in De’an County, northern Jiangxi Province. The simulated rainfall hall covered an area of 1200 m^2^, and the top was equipped with a combined downward spray artificial rainfall device. During simulation, the rainfall intensity was managed by adjusting the water pressure in the supply pipe to ensure the homogeneous distribution and stable velocity of raindrops for making the simulations as close to natural rainfall as possible. The fall height of simulated raindrops was 14 m to ensure that all raindrops could reach the terminal velocity of natural raindrops and a uniformity ratio of 90%. The rainfall intensity could be adjusted in the range of 10 to 220 mm h^−1^ by changing the nozzle size and the water pressure. 

Before the formal experiment, the soil surface was pre-wetted at a rainfall intensity of 10 mm h^−1^ until the bottom penetration water appeared. After the soil plot was left horizontally for 12 h, the simulated rainfall was started.

According to the multi-year rainfall data and long-term meteorological observation data in the study area, the rainfall simulations in this study were set as three rainfall intensities, 45, 90, and 135 mm h^−1^, for a duration of 90 min. These simulated rainfall settings represent typical rainfall regimes in the granitic-parent-haired red soil region of southeastern China [13].

### 2.4. Data Collection and Analysis

During each simulation, the initial time of the surface and subsurface flows, as well as the bottom penetration, were recorded. Subsequently, the flow and sediment samples were collected at 3 min intervals, and the total collecting time was 90 min for surface flow and sediment samples. Considering the trailing of subsurface flow and bottom penetration, the subsurface flow and bottom penetration were continued to be collected at 3 intervals for a total collection time of 240 min. The sediment samples were dried in a forced-air oven at 105 °C until a constant mass was obtained and weighed to calculate the sediment concentration and soil loss rate.

After uniformly mixing all surface flow and sediment samples collected during simulation, five bottles of them were taken to the laboratory for analysis of sediment particle size. The sediment particle size was measured by the laser particle size analyzer (model Eye tech) produced by Amide Company; 0–12,000 μm of particle size distribution was selected. We calculated the corresponding particle sizes D10, D50, and D90 when the cumulative particle size distribution of sediment reached 10%, 50%, and 90%, respectively. After each rainfall simulation, the sediments were collected from the upper and lower parts of the soil slope with a thickness of 2 cm near the surface and were brought back to the laboratory immediately for particle size sieving. In this study, the air-dried soil samples were passed through a set of sieves including 2, 1, 0.5, 0.25, and 0.15 mm to determine the weight distribution of different particle sizes, followed by oven drying to obtain their mass. 

The surface flow coefficient (*SFC*) was determined as the flow depth per minute to plot the area with the following formula:(1)$$SFC=V_{S_{f}}/V_{r}$$
where SFC is the surface flow coefficient (%), $V_{S_{f}}\;$ is the surface flow volume (L), and $V_{r}$ is the rainfall volume (L).

The sediment concentration (*SC*) of each sample was determined using the following formula:(2)$$SC=W/V_{S_{f}}$$
where *SC* is the sediment concentration (g L^−1^), and *W* is the sediment weight of each sample (g). 

In this study, the rainfall transformation ratio (*RTR*) was computed according the following equation:(3)$$RTR=\left(V_{f}/V_{r}\right)\times 100\%$$
where RTR is the rainwater transformation ratio to surface and subsurface flow, as well as the bottom penetration, represented as a non-dimensional quantity, and $V_{f}$ represents the amount of surface flow, or subsurface flow, or bottom penetration; the unit of measure for all variables is L.

Microsoft Excel 2019 and SPSS 25.0 were used to organize and statistically analyze the above experimental data. The differences among the treatments were tested by using one-way ANOVA at the *p* = 0.05 probability level.

## 3. Results

### 3.1. Surface Flow Generation

Figure 3 presents the surface flow coefficients under different rainfall intensities. When the rainfall time increased, the surface flow increased first and then hit a relatively stable value; therefore, the surface flow coefficient showed an increasing–stabilizing tendency. The response of surface flow accelerated with an increase in rainfall intensity. When the rainfall intensity increased from 45 to 135 mm h^−1^, the initial generation time was decreased by 4 min (Table 2). Moreover, the time to reach the steady state was 36 min under 45 mm h^−1^ rainfall and was shortened to 24 and 15 min under 90 and 135 mm h^−1^ rainfall events, respectively.

The surface flow coefficients increased significantly with increasing rainfall intensities. At 45 mm h^−1^ rainfall, the initial surface flow coefficient was only 1.95%, while the values were 32.49% and 55.97% at 90 and 135 mm h^−1^ rainfall intensities, respectively. Moreover, the peak value also increased from 27.36% under 45 mm h^−1^ to 97.13% under 135 mm h^−1^ rainfall. For the mean surface flow coefficient, it was only 19.87% at 45 mm h^−1^ rainfall and increased to 83.84% and 92.42% at 90 and 135 mm h^−1^ rainfall intensities, respectively.

### 3.2. Subsurface Flow Generation

The generation of subsurface flow showed an increasing–stabilizing–decreasing trend (Figure 4). All subsurface flow amounts increased sharply to a high value, subsequently reached a relatively stable stage, and then declined rapidly after rainfall stopped. The initial generation of subsurface flow lagged behind the surface flow, and the lag time ranged from 2 to 7 min (Table 2). However, the time to reach the stable state increased with an increase in rainfall intensity, which was from 40 min at 45 mm h^−1^ increased to 70 and 55 min at 90 and 135 mm h^−1^. 

The initial subsurface flow generation increased with increasing rainfall intensities. At a 45 mm h^−1^ rainfall event, the initial flow amount was 26 mL, which increased to 43 and 88 mL at 90 and 135 mm h^−1^ rainfall intensities, respectively. With the extension of rainfall time, the generation of subsurface flow had a negative relationship with rainfall intensity. When the rainfall intensity rose from 45 to 135 mm h^−1^, the peak value decreased from 270 to 210 mL. Moreover, during the whole rainfall period, the total amount at 45 mm h^−1^ was 1.1 L higher than that at 90 and 135 mm h^−1^ rainfall.

After rainfall stopped, the subsurface flow continued to generate, and the flow amount was even more than that during the rain. At the 45 mm h^−1^ rainfall event, the subsurface flow amount after the rain was 8.3 L, which was 2.1 L greater than that during the rain process. For the 90 and 135 mm h^−1^ rainfalls, the post-rain value was approximately 3 L greater than that the in-rain value.

### 3.3. Bottom Penetration

The generation of bottom penetration presented an increasing–decreasing tendency, with one peak (Figure 5). All bottom penetration increased sharply to a peak and then decreased gently. The initial generation time and peak time extended greatly with increasing rainfall intensity. From 45 to 135 mm h^−1^ rainfall, the initial time increased from 13 to 25 min, and the peak time increased from 87 to 99 min. The higher the rainfall intensity, the slower the bottom penetration response. 

The bottom penetration generation was negatively correlated with rainfall intensity (Table 2). With the increase in rain intensity, the peak amount decreased from 470 to 341, and to 290 mL. During the entire rainfall process, the total bottom penetration amount significantly decreased from 7.4 L at 45 mm h^−1^ rainfall to 4.8 and 3 L at 90 and 135 mm h^−1^, respectively. After the rainfall stopped, the bottom penetration amount was 8.4 L under 45 mm h^−1^ rainfall intensity, which was 2.3 and 1.8 L higher than that at 90 and 135 mm h^−1^, respectively.

### 3.4. Sediment Production

The variability of sediment concentration with flow time under different rainfall intensities is depicted in Figure 6. Under all rainfall intensities, the sediment production process showed a decreasing–stabilizing trend, with dynamic fluctuations. The sediment concentration decreased rapidly within 15 min after the onset of rainfall, and then the decrease became smaller. The time to reach the stable state advanced with the increasement of rainfall intensity. At 45 mm h^−1^ rainfall intensity, this time was 63 min, which shortened to 42 and 30 min at 90 and 135 mm h^−1^ rainfall intensities, respectively.

The sediment concentrations increased significantly with increasing rainfall intensities. At 45 mm h^−1^ rainfall, the initial sediment concentration was only 1.39 g L^−1^, which was greatly lower than 12.59 and 21.16 g L^−1^ at 90 and 135 mm h^−1^, respectively. Moreover, for the average sediment concentration, this value increased from 0.62 g L^−1^ at 45 mm h^−1^ to 4.80 and 7.70 g L^−1^ under 90 and 135 mm h^−1^ rainfall events, respectively. 

Three indices, namely, D_10_, D_50_, and D_90_, which were used to characterize the particle size, all increased with an increase in rainfall intensity (Figure 7). At 45 mm h^−1^ rain, D_10_ was 30.51 μm and increased to 41.22 and 45.75 μm at 90 and 135 mm h^−1^ rainfall intensities, respectively. The increases in D_50_ and D_90_ were more significantly affected by the rainfall intensity. From 45 to 90 and 135 mm h^−1^ rainfall, D_50_ increased by 1.61 and 1.83 times, respectively. Moreover, for D_90_, this value was 49.15 and 80.13 at 135 mm h^−1^ more than that at 90 and 45 mm h^−1^ rainfall, respectively.

## 4. Discussion

### 4.1. Variations in Particle-Size Distribution of the Eroded Surface Soil

Variations in PSD can reflect the effects of soil properties such as soil texture, aggregate content [40], and soil moisture [41], as well as rainfall characteristics such as rainfall intensity, duration [42], and rainfall redistribution including surface and subsurface flow [43,44] on soil erosion processes. 

A 2 cm thick sediment layer on the surface was collected after rainfall ended for sieving analysis to determine the variations in PSD (Figure 8). The results showed that with an increase in rainfall intensity, the contents of large (>2 mm) and medium (1–2 mm) particles of the eroded sediment were increased. When the rainfall intensity was raised from 45 to 90 and 135 mm h^−1^, the content of particle with size (>1 mm) increased from 51.8% to 61.6% and 63.8%, respectively. Conversely, all particles with sizes (<1 mm) significantly decreased with the increase in rainfall intensity. The values were reduced from 48.2% to 38.4% and 36.2%, respectively.

Ultimately, under light rain, the content of sand particle (>0.5 mm) decreased by only 3.25% from the original value of 72.55%, while it increased by about 6% under heavy rains. The content of fine particles (silt and clay) increased from the original 27.44% to 30.7% at 45 mm h^−1^ rainfall and significantly decreased to 21.1% and 21.9% at 90 and 135 mm h^−1^, respectively. The stronger rainfall intensity, the more enrichment of coarse particles and loss of fine particles on the soil surface, leading to the degradation of soil quality and fertility [45]. This is due to the fact that at lower rainfall intensities, coarse particles are dispersed into smaller particles by the kinetic energy of raindrops, and small particles are lost in suspension, while large particles are harder to be transported. As the transportable fine particles on the soil surface are exhausted, the coarser particles are exposed in the topsoil layer to protect the underlying soil. With the increase in rainfall duration, the content of fine particles gradually decreases, while the content of coarse particles gradually increases. Under larger rainfall intensities, there is the increase in stream power and carrying capacity of the surface flow. Fine particles are transported by suspension–saltation, and coarse particles can also be transported, which obviously increases the proportion of coarse particles in the eroded surface soil.

Variations in PSD indicate that surface soil particles developed from weathered granite are easily sorted under the disturbance of rainfall, finer particles including clay and silt deposit at the bottom, and coarser sand particles gradually cover the surface. This changes the composition and structure of soil [46] and has significant effects on the rainfall–flow redistribution, processes of flow generation, and sediment production (Figure 8).

### 4.2. Characteristics of Rainfall–Flow Redistribution

The infiltration process was significantly affected by soil texture and rainfall intensity, including infiltration amount, rate, and depth on the soil surface, which was an important reason for the differences in rainfall redistribution. It in turn affected the surface–subsurface flow generation process. Coarsened soil significantly promoted the infiltration process under light rainfall intensities (Figure 9). The transformation ratio of rainfall into surface flow was only 25.19%; the other three-quarters of rainwater infiltrated downwards; and 35.80% and 39.01% were converted into subsurface flow and flowed out through bottom penetration, respectively. Subsurface flow and bottom penetration were the main output forms of rainwater in sloping lands. Coarsened soil showed a strong infiltration–promoting capacity but with a poor water-storage capacity under light rainfall intensities.

When the rainfall intensity increased, the rainfall rate exceeded the infiltration rate. At this time, the infiltration process was mainly controlled by soil conditions including soil structure, soil texture, and soil particle size distribution. The part of rainwater that exceeded the infiltration rate immediately transformed to the surface flow, and thus the mode of flow generation changed from saturation excess flow to infiltration excess flow [47]. Most rainwater was exported in the form of surface flow. At 90 mm h^−1^ rainfall, the transformation ratios of rainfall into surface and subsurface flow were 78.87% and 11.53%, respectively, and the bottom penetration was only 9.60%. Such a transformation difference became more significant with an increase in rainfall intensity. When the rainfall intensity rose to 135 mm h^−1^, the conversion rate of surface flow increased to 87.07%, while the value of subsurface flow and bottom penetration decreased to only 7.48% and 5.45%, respectively. Therefore, in the case of heavy or extreme rainfall, the increased loss of rainwater in the form of surface flow exacerbated the scouring effect on the soil surface. Coarsening soil surface and massive loss of fine particles resulted in the degradation of soil equality and fertility [48,49,50]. 

### 4.3. Characteristics of Flow Generation

The characteristics of weathered granite red soil cause the flow generation time of different soil layers to be significantly different [51,52,53,54]. In the early stage of rainfall, the rainwater is mainly used to moisten the surface soil and fill the soil pores, and thus the flow generation has a hysteresis effect. Before reaching soil saturation, the surface flow is generated in the form of infiltration excess flow [47]. Under light rainfall intensity, due to the rough texture, large pores, and poor water holding capacity of weathered granite red soil [55], the rainwater is more likely to infiltrate downwards. Therefore, the surface flow generation is slower, and the subsurface flow and bottom penetration generations are faster. After the surface soil reaches saturation, the form of surface flow generation transforms to saturated flow [47,56]. At this time, the subsurface flow generation reaches a relatively stable stage. The stronger the rainfall intensity, the more rainwater enters the soil plot per unit time, and the faster the soil reaches saturation as well as the surface flow generation [57,58]. 

The flow generation process during rainfall is affected by the change of the soil environment of weathered granite. In the early stage of rainfall, under the splashing and compact effect of raindrops, soil aggregates are destroyed and formed to silt and clay particles, blocking soil pores and forming thin soil crusts [59]. Soil crusts reduce the surface roughness and infiltration, which is beneficial for the surface flow generation and reduce subsurface flow and bottom penetration generation [60,61]. However, with the extension of rainfall, coarse particles gradually cover the soil surface, which reduces the formation of soil crusts and increases porosity to promote rainwater and flow infiltration, causing fluctuations in the process of flow generation [37,56].

### 4.4. Prediction of Subsurface Flow and Bottom Penetration Generation Processes

In the weathered granite red soil sloping farmland, the coarse soil texture leads to strong soil permeability but poor water storage capacity; the migration of flow and sediment of different soil layers is also complex. The infiltration process and near-surface hydrology significantly affect the soil erosion process and are related to nutrient loss and pollutant migration in the agricultural lands. Therefore, the prediction of near-surface hydrological processes is particularly important.

When the infiltration process is mainly controlled by soil conditions rather than rainfall intensity, it is a function of rainfall time. With the extension of rainfall time, subsurface flow and bottom penetration can be predicted by the one-dimensional quadratic polynomial of the rainfall time (Figure 10), and the relationship is expressed as follows:(4)$$V=AT^{2}+BT+C$$
where V is the amount of subsurface flow or the amount of bottom penetration (L); T is the rainfall time (min); and *A*, *B*, and *C* are constants. 

The coefficients of determination of the subsurface flow and bottom penetration fitting with rainfall all increase with an increase in rainfall intensity. Moreover, the fitting relationship of the bottom penetration and rainfall time is better than that of the subsurface flow (Figure 10). 

Considering that after rainfall stops, the subsurface flow and bottom penetration continue to generate, and the total amount even exceeds that during the rainfall. Therefore, the prediction of subsurface flow and bottom penetration after rainfall stopped is more important.

After rainfall ends, the fading process of subsurface flow and bottom penetration is expressed by an exponential function relation of time (Figure 11). The longer the time, the lower the subsurface flow and bottom penetration. The exponential relationship is as follows:(5)$$v=e^{at+b}$$
where v is the amount of subsurface flow or the amount of bottom penetration after the end of rainfall (L), t is the time after rain stopped (min), and a and b are constants.

With an increase in rainfall intensity, the subsurface flow and bottom penetration fit better with rainfall time. The fitting relationship between bottom penetration and rainfall time (*R*^2^ > 0.98) is better than that of the subsurface flow (*R*^2^ > 0.92) (Figure 11).

Figure 12 presents the relationship between the measured and predicted values of subsurface flow and bottom penetration after rainfall ends. The predicted and measured values were in good agreement with the 1:1 line, which indicates that the subsurface flow and bottom penetration can be predicted well. As time continues, the predicted values were gradually lower than the measured values, especially for the subsurface flow. In general, with an increase in rainfall intensity, the coefficients of determination of the prediction equations of subsurface flow and bottom penetration all increased. The stronger the rainfall intensity, the higher the prediction accuracy. Moreover, the coefficients of determination of bottom penetration were generally higher than that of subsurface flow, meaning that the prediction accuracy of bottom penetration was higher than that of subsurface flow.

### 4.5. Characteristics of Sediment Production

Compared with flow generation, the sediment production process is more fluctuated. The reason was that the processes of slope erosion and sediment transport were unsaturated and non-equilibrium. The erosion capacity of surface flow and the supply of erosive materials on the slope together determined the sediment production. In all rainfall events, the sediment concentration firstly rapid decline from a high value, accompanied by dynamic fluctuations. This can be partly attributed to the first washout effect of the erosion process [55,57,58,62]. Moreover, due to the soil surface developed from the weathered granite parent material being loose, a large number of soil particles are dispersed by the splashing effect of raindrops. More loose sediment can be carried by surface flow, resulting in a high sediment concentration, while the subsequent decrease in sediment concentration is due to the reduction of mobile soil particles, as well as the increase in soil stability under the compaction effect of raindrops and the transient formation of thin surface crusts [59,63]. Meanwhile, the increased value of surface flow also led to the decreased value of sediment concentration. In the middle and late stages of rainfall, the separation capability of surface flow under the disturbance of raindrops and the anti-erodibility of soil surface reached a balance, causing the variation of sediment production to tend to be stable.

Rainfall directly affects sediment production through the impact of raindrops and indirectly affects it through the differences in rainfall–flow redistribution. Under light rain, the dispersed sediment particles are less under the lower kinetic energy of the raindrops and less surface flow. Moreover, due to the loose texture and strong permeability of the granite weathered red soil [64], the infiltration volume and rate are greater and faster, which slows down the surface flow amount and rate and decreases the flow energy. This leads to less soil loss. While under greater intensity rainfall events, the higher kinetic energy of raindrops separates more sediment particles from the loose surface, which becomes the major source of soil transportation and loss. In addition, the more rainwater transformed into surface flow, the flow velocity and energy increase under heavier rainfall, which increases the ability of surface flow to separate and transport sediment. Vaezi et al. [65] pointed out that a high initial value of sediment concentration was associated with an increase in surface flow to separate soil particles.

However, with the extension of rainfall, covered coarse particles weakens raindrop splash erosion of the surface soil [66], reduces the physical degradation (such as surface sealing and compaction) of the soil surface [67], and enhances the anti-erodibility and anti-scourability of the soil [68]. In addition, the existence of coarse particles may increase porosity to increase flow infiltration, and thus the detachment and transport capability of the surface flow is decreased [69], reducing soil loss [70,71]. This can explain that sediment production enters a relatively stable state in the later period of rainfall.

## 5. Conclusions

Simulated rainfall experiments were conducted to study the processes of rainfall-flow redistribution, flow generation and prediction, and sediment production in red soil sloping farmland developed from weathered granite. Under light rainfall intensities, the subsurface flow and bottom penetration were the main output of rainwater, and coarsened soil exhibited strong infiltration-promoting but poor water-storage capacities. With an increase in rainfall intensity, the surface flow generation and sediment production significantly increased. Moreover, the content of coarse particles (>1 mm) in surface soil increased by 12% while the content of fine particles (<0.5 mm) decreased by 9.6%. This led to severe coarsening of surface soil, as well as degradation of soil equality and fertility under heavy or extreme rainfall intensities. The prediction of the near-surface hydrological process was important for preventing water erosion and rainwater utilization. During rainfall simulation, the subsurface flow and bottom penetration were predicted through a quadratic equation about rainfall time; after rainfall ended, the prediction equations were transformed into time-dependent exponential functions. The prediction accuracy increased with an increase in rainfall intensity, and the prediction accuracy of bottom penetration was higher than that of subsurface flow. 

The results suggested that rainfall–flow redistribution and near-surface hydrological processes had to be taken into account in coarse-textured agricultural lands in the hilly area, as well as the fact that increasing surface cover is the key to prevent soil erosion and regulate water resources on the severely eroded bared land.

In situ simulated experiments on granite red soil slopes should be conducted to verify and supplement the indoor experiments. The variations of hydraulic properties of flow should be analyzed in depth, and the hydrodynamic characteristics and erosion mechanism of granite red soil should be further explored.

## Figures and Tables

**Figure 1 ijerph-20-02104-f001:**
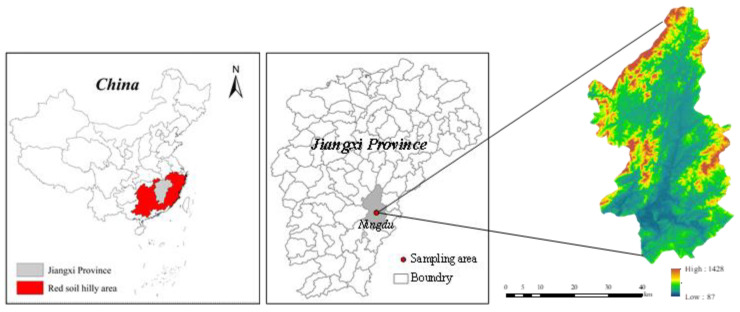
Location of study area.

**Figure 2 ijerph-20-02104-f002:**
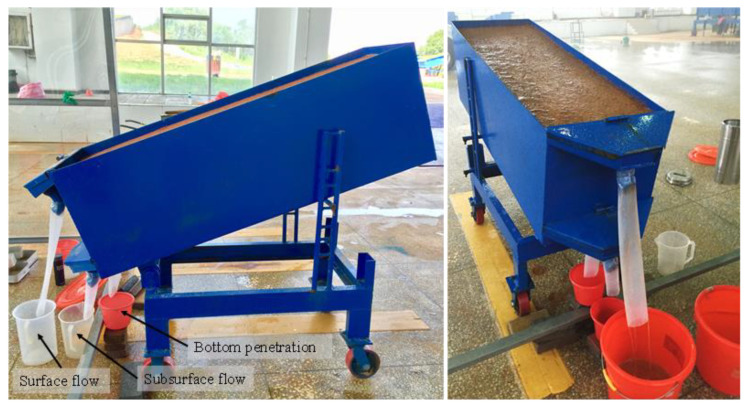
The structure of the experimental soil plot.

**Figure 3 ijerph-20-02104-f003:**
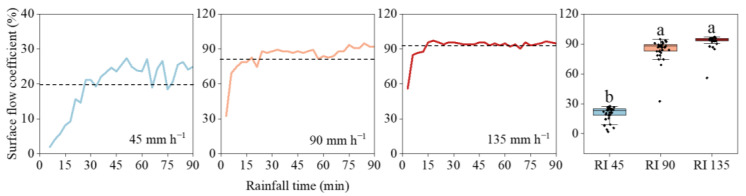
The surface flow generation processes under different rainfall intensities. The dotted lines indicate the mean surface flow coefficients. Error bars represent standard errors of the means (*n* = 30). Bars with different letters differ significantly.

**Figure 4 ijerph-20-02104-f004:**
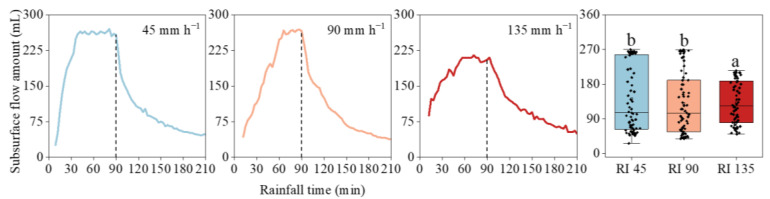
The subsurface flow generation processes under different rainfall intensities. Error bars represent standard errors of the means (*n* = 80). Bars with different letters differ significantly.

**Figure 5 ijerph-20-02104-f005:**
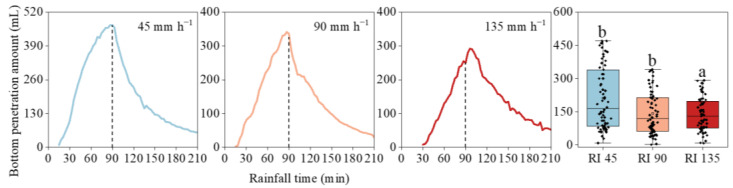
The bottom penetration generation processes under different rainfall intensities. Error bars represent standard errors of the means (*n* = 80). Bars with different letters differ significantly.

**Figure 6 ijerph-20-02104-f006:**
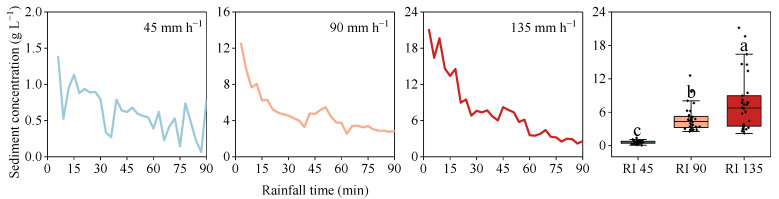
The sediment concentration under different rainfall intensities. Error bars represent standard errors of the means (*n* = 30). Bars with different letters differ significantly.

**Figure 7 ijerph-20-02104-f007:**
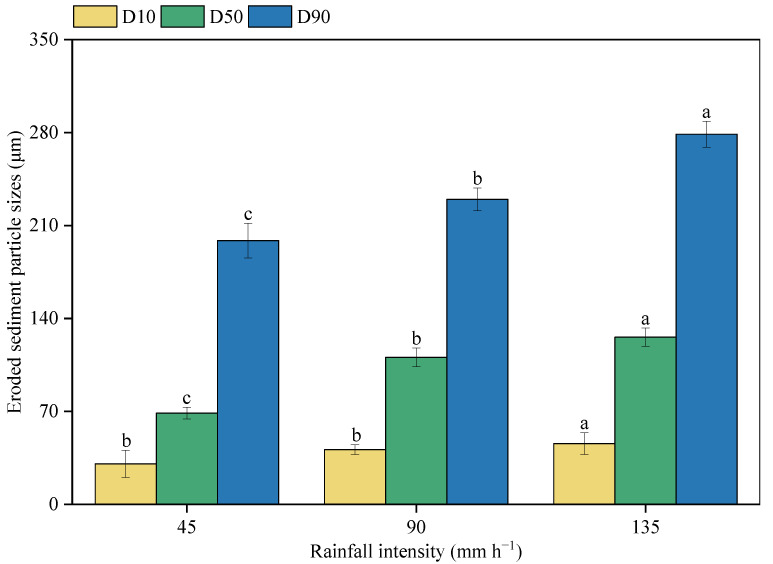
The size distributions of the eroded sediments under different rainfall intensities. Note: D10, D50, and D90 were the corresponding particle sizes when the cumulative particle size distribution of the sediment samples reached 10%, 50%, and 90%. The different letters indicate significant differences among different rainfall intensities (*p* < 0.05).

**Figure 8 ijerph-20-02104-f008:**
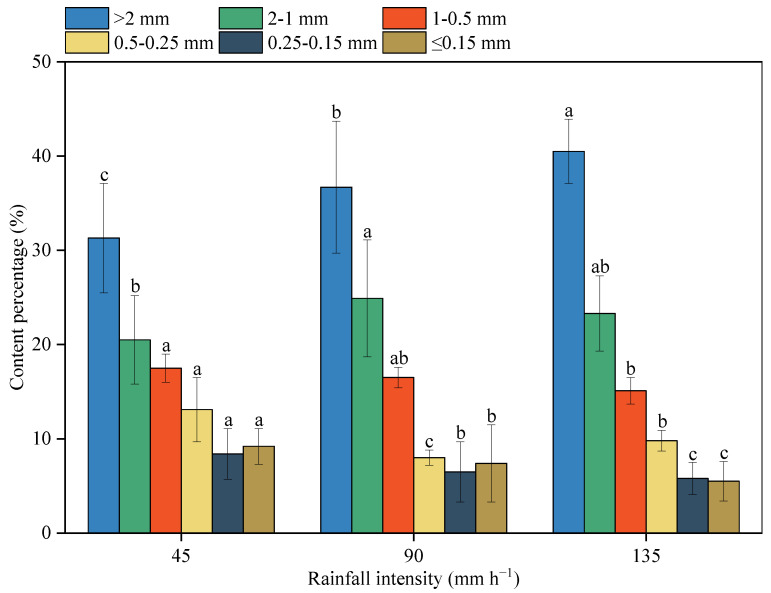
The sieved size and weight percentage of the eroded surface soil after rainfall simulation. The different letters indicate significant differences among different rainfall intensities (*p* < 0.05).

**Figure 9 ijerph-20-02104-f009:**
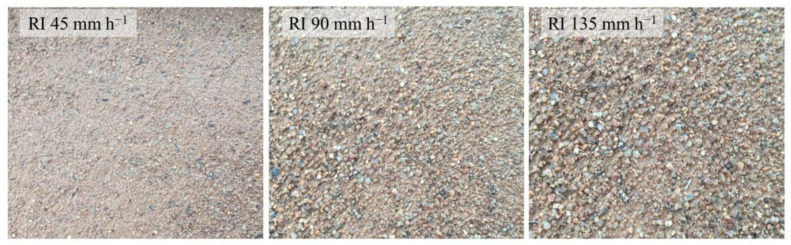
The coarsen feature of the soil surface under different rainfall intensities.

**Figure 10 ijerph-20-02104-f010:**
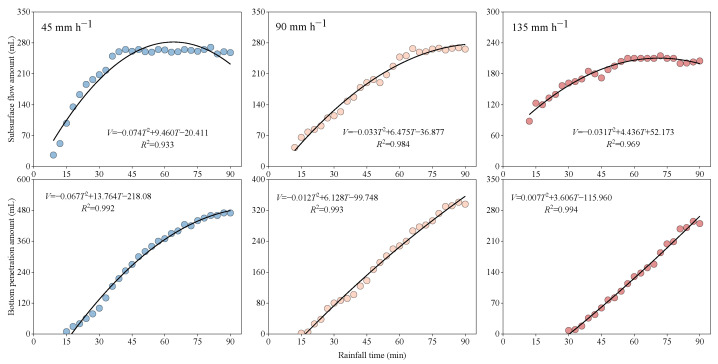
Relationships between the times and subsurface flow and bottom penetration during the rainfall simulations.

**Figure 11 ijerph-20-02104-f011:**
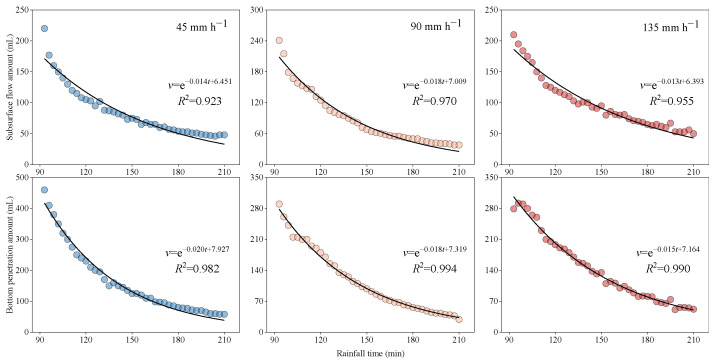
Relationships between the times and subsurface flow and bottom penetration after the rainfall simulations ends.

**Figure 12 ijerph-20-02104-f012:**
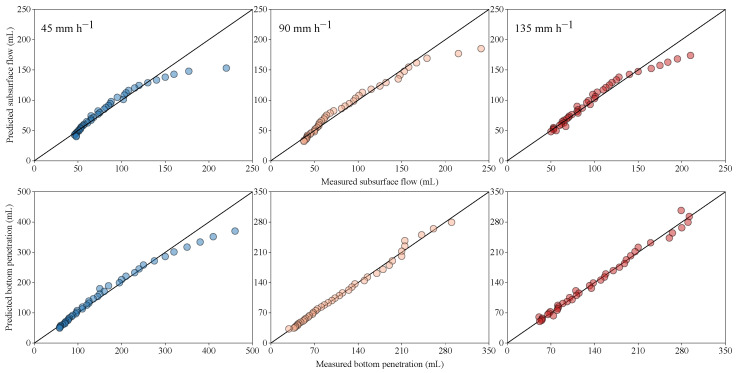
Relationships between the measured and predicted subsurface flow and bottom penetration after rainfall simulations.

**Table 1 ijerph-20-02104-t001:** Characteristics of the soil used in the present study.

Parameter	Unit	Value
Coarse sand (200–2000 μm)	g kg^−1^	486.57 ± 22.62
Fine sand (50–200 μm)	g kg^−1^	238.86 ± 18.74
Coarse silt (20–50 μm)	g kg^−1^	43.52 ± 11.66
Fine silt (2–20 μm)	g kg^−1^	76.51 ± 6.90
Clay (0–2 μm)	g kg^−1^	154.54 ± 29.0
Organic matter	g kg^−1^	5.29 ± 0.97
TN	g kg^−1^	0.37 ± 0.08
TP	g kg^−1^	0.12 ± 0.03
CEC	mol kg^−1^	11.91 ± 0.96
pH		4.65 ± 2.91

TN, total nitrogen; TP, total phosphorous; CEC, cation exchange capacity.

**Table 2 ijerph-20-02104-t002:** The hydrological characteristics of the rainfall–flow under different intensities.

Rainfall Intensity (mm h^−1^)	Initial Generation Time (min)			Peak Flow Amount (L)			Total Flow Amount (L)		
	Surface Flow	Subsurface Flow	Bottom Penetration	Surface Flow	Subsurface Flow	Bottom Penetration	Surface Flow	Subsurface Flow	Bottom Penetration
45	5	7	13	0.51	0.27	0.47	10.2	14.5	15.8
90	2	9	15	3.50	0.27	0.34	89.6	13.1	10.9
135	1	7	25	5.38	0.21	0.29	153.6	13.2	9.6

## Data Availability

Not applicable.

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
