# Peer review of "Erosion of Granite Red Soil Slope and Processes of Subsurface Flow Generation, Prediction, and Simulation"

_ijerph, 2023, doi:10.3390/ijerph20032104_

Round 1

Reviewer 1 Report

Background:

In this study, on the red soil slope farmland developed from weathered granite, simulated rainfall experiment was conducted to study the characteristics of rainfall redistribution, the processes of surface–subsurface flow generation and prediction, and sediment production. Rainfalls with three intensities of 45-, 90-, and 135-mm h-1 with a duration of 90 minutes were applied to the weathered granite red soil with the slope gradient of 10°. Under 45 mm h-1 rainfall intensity, the output of rainwater was composed by subsurface flow and bottom penetration, accounting for 35.80% and 39.01% of total rainfall, respectively.

General comment: This is an important topic relevant to the journal and to the readership. The methodology is sound, the results are generalizable, and add significantly to what is already known about this topic. There are no problems with ethics or conflict of interest."

Introduction:

I have no comments; very well written, and in the introduction needed to add the recent references.

Materials and Method:

1- Do you measure the effect of organic carbon contents soil holding capacity? (It is a very important soil parameter)

2- What are the main chemical characteristics of the soil?

3-Do the authors need to explain why they choose the selected soil type?

Results:

1-The figures are too much (12 fig).

2- Please add the following sentence under the figures.

Error bars represent standard errors of the means (n = …..). Bars with different letters differ significantly.”

Discussion:

very well written.

Conclusion

In your conclusion, please discuss limitations of the present study and future research to address them.

Reviewer 2 Report

It should be noted that the manuscript's topic is interesting. 

In introduction you should have used the literature review base on your topic.

What is the novelty of your research?

Line 104. In this section it is better to show the location of case study by a map and map of geology and map of soil. According to your topic and methods, I suggest you, discuss about the soil texture. 

Line 147. Why 1200 m-2?

Line 200-202 should be in materials and methods.

Line 233. What is mL?

Line 234. Do you mean rainfall intensity?

Line 334. What about soil texture?

Line 399. Reference?

Line 405. In this section, you can use RMSE.

In discussion, talk about the detail of soil texture of the case study. 

Reviewer 3 Report

1) Please note the superscript in the unit (e.g., Lines 13-14,17 mm h-1, Line 22 g L-1, Line 59 t ha-1 yr-1, Line 147 1200m2, not 1200m -2).

2) In the introduction, we should add the literature of the past five years to better clarify the current situation and existing problems of the study, as well as the significance of this study.

3) Lines 121-122: The soil texture belongs to the loamy sandy soil, with the sand content of 72.55%, the silt content of 12.38%, and the clay content of 15.06%. Lines 340-342: When the rainfall intensity raised to 135 mm h-1. The conversion rate of surface flow increased to 87.07%, while the value of subsurface flow and bottom penetration decreased to only 7.48% and 5.44%, respectively. Why do they all add up to 99.99% instead of 100%? Is there anything else?

4) Lines 159-161: Why these three kinds of rain intensities are relatively typical should be supported by supplementary literature.

5) Is there any difference between Vr in lines 185-187 and Rrain in lines 195-199?

6) It is suggested to draw the mean surface flow coefficient line with dotted lines in the first three figures of Figure 2.

7) Since the abscissa of the last figure in Figure 2, Figure 3 and Figure 4 is different from that of the first three, it is recommended to distinguish their abscissa.

8) It is suggested to add a table to list the subsurface surface flow amount and bottom penetration amount during and after rainfall. You can also consider redesigning this table and Table 1.

9)Line 382: V is the subsurface flow amount or the bottom penetration amount during rainfall rather than the sum of them. Lines 400-401: v is the amount of subsurface flow or bottom penetration after the end of rainfall.
